

# Scan strategies for wind profiling with Doppler lidar - An LES-based evaluation.

Charlotte Rahlves[1], Frank Beyrich[2], and Siegfried Raasch[1]

[1]Leibniz University Hannover, Institute of Meteorology and Climatology, Hannover, Germany
[2]Meteorological Observatory Lindenberg, Richard-Aßmann-Observatory, German Meteorological Service, Germany

**Correspondence:** Charlotte Rahlves (rahlves@muk.uni-hannover.de)

**Abstract.**

Lidar scan techniques for wind profiling rely on the assumption of a horizontally homogeneous wind field and stationarity for the duration of the scan. As this condition is mostly violated in reality, detailed knowledge of the resulting measurement error is required. The objective of this study is to quantify and compare the expected error associated with Doppler-lidar wind
profiling for different scan strategies and meteorological conditions by performing virtual measurements implemented in a large-eddy simulation (LES) model. Various factors influencing the lidar retrieval error are analyzed through comparison of the wind measured by the virtual lidar with the 'true' value generated by the LES. These factors include averaging interval length, zenith angle configuration, scan technique and instrument orientation. For the first time, ensemble simulations are used to determine the statistically expected uncertainty of the lidar error. The analysis reveals a root-mean-square deviation
(RMSD) of less than $1\,\mathrm{m\,s^{-1}}$ for 10 min averages of wind speed measurements in a moderately convective boundary layer, while RMSD exceeds $2\,\mathrm{m\,s^{-1}}$ in strongly convective conditions. Unlike instrument orientation and scanning scheme, the zenith angle configuration proved to have significant effect on the retrieval error. Horizontal wind speed error is reduced when a larger zenith angle configuration is used, but is increased for measurements of vertical wind. Results suggest that the scan strategy has a relevant effect on the lidar retrieval error and that instrument configuration should be chosen depending on the quantity
of interest and the flow conditions in which the measurement is performed.

## 1 Introduction

Profiling Doppler lidars are nowadays widely used for applications like wind energy, airport safety and air quality control (Courtney et al., 2008; Antoniou et al., 2007; Emeis et al., 2007; Nechaj et al., 2019; Cottle et al., 2014). Recently, lidars have become relevant in the field of numerical weather prediction (NWP). State-of-the-art NWP models increasingly require
wind profile measurement data of the atmospheric boundary layer (ABL) for assimilation (Knist et al., 2018), to improve the prescription of initial conditions for the simulations. Profiling lidars with a reasonably high vertical resolution can cover almost the entire vertical extent of the ABL, except for the lowest 100 m. However, lidar scan techniques rely on the assumption of a horizontally homogeneous wind field and stationary conditions during the measurement, since a series of wind measurements is performed sequentially along slanted paths at different azimuth directions. These assumptions are rarely fulfilled in reality





due to turbulent fluctuations of the wind field, especially in the convective boundary layer. Therefore, detailed knowledge of the resulting retrieval error is required for a realistic estimate of the uncertainty that is to expected from lidar wind measurements.

Validation of lidar-based wind data against established instruments for wind measurements, such as mast-mounted anemometers, radiosondes or radar wind profilers have been carried out by a number of studies (Smith et al., 2006; Kindler et al., 2007; Gottschall et al., 2012; Päschke et al., 2015). However, such validation experiments are not straightforward to interpret because of range limitations, different vertical resolution, and different measurement and sampling principles. Here, Computational Fluid Dynamics (CFD) can serve as an alternative method to assess the uncertainties of lidar-based wind retrievals. When performing a simulation of the flow, the flow variables are precisely known at all points of the numerical grid. A 'virtual' instrument can then be implemented into the simulation to 'measure' the wind field in the same manner a real lidar does. This virtual measurement can then be compared to the 'true' wind field, as 'known' by the simulation. Thus, the retrieval error associated with a specific sampling strategy can be exactly determined from the deviation. Large-eddy simulations (LES) are particularly suitable for this task, as they explicitly resolve the bulk of turbulent motions and offer the possibility for high grid resolution. Recent studies have used this approach to investigate the errors occurring in lidar measurements due to turbulence induced heterogeneity of the flow (Lundquist et al., 2015; Gasch et al., 2020; Stawiarski et al., 2015). However, these studies either feature specific flow conditions, such as wake flows behind wind turbines (Lundquist et al., 2015), or employ distinct scanning methods, such like an airborne lidar (Gasch et al., 2020) or a Dual-Doppler lidar (Stawiarski et al., 2015). Also, usually only one single scanning scheme has been employed and only individual simulation runs were conducted to determine the lidar retrieval error.

Based on the need for practical guidelines for operational use, we take a more general approach. Focusing on widely used scan strategies for ground based lidars, we investigate various factors that may influence the lidar error. Those factors include the scanning scheme used for wind vector retrieval, the number of beams employed during one scan cycle, the orientation of the lidar with respect to the prevailing wind direction and the zenith angle configuration of the instrument. In addition, measurements averaged over different time interval lengths will be analyzed to answer how optimal application of temporal averaging can mitigate lidar measurement inaccuracies. Coherent flow features that are often observed in the ABL violate the basic assumption of a horizontally homogeneous wind field. Since the formation of such features as cellular structures, streaks or roll convection is influenced by the atmospheric stability and the mean vertical wind shear (Deardorff, 1972; Moeng and Sullivan, 1994; Khanna and Brasseur, 1998; Salesky et al., 2017), accuracy of lidar measurements will also depend on the respective atmospheric conditions. Therefore, virtual lidar measurements will be assessed for various convective flow regimes, distinguished by their relation of buoyancy to shear, with the goal of better understanding the error behavior for each specific regime. The main objective of this study is to identify the most advantageous scan strategy that minimizes the lidar retrieval error for each regime and to provide a reliable estimate for this error.

We conduct several simulations of lidar measurements in atmospheric boundary layers with different geostrophic forcing and stratification. Wind measurements with virtual lidars are performed using different scanning schemes and zenith angles simultaneously. The employed scanning schemes include velocity azimuth display (VAD) with 6 and 24 scanning beams, as well as a Doppler-beam swinging (DBS) scheme using four beams. We select three zenith angle configurations, of which two



(15° and 30°) are commonly applied in practice. The third configuration of 54.7° is chosen, based on a study by Teschke and Lehmann (2017) who found that this angle minimizes the effect of error propagation of radial wind measurement errors on the retrieved wind vector. Moreover, this comparably large zenith angle is recommended for Dopple lidar operation if one additonally wants to derive turbulence variables from the scans (e.g. Smalikho and Banakh (2017)). The resulting wind profiles are compared with the true profile above the lidar location (reference profile). To quantify the deviation from reference we

apply the RMSD as a mean over the vertical extent of the boundary layer. This way we evaluate the lidar error related to each scan strategy and identify the optimal scan strategy for each meteorological situation. For two simulation cases we extend the individual simulations to an ensemble of ten simulations each to increase the statistical confidence of the lidar errors found.

This study is structured as follows. Section 2 describes the lidar scanning schemes that were used for the virtual lidar measurements. Section 3 explains the operating procedure of the virtual lidar simulator, including the implementation of the

different scanning schemes, and provides the metric for error quantification. Section 4 outlines the simulation setups. We present the results of the lidar simulations in section 5. In section 6 we finally discuss the results and their implications and give some recommendations for further studies that we suggest to perform based on our results.

## 2 Lidar scanning schemes

Monostatic pulsed Doppler lidars make use of the frequency shift between an emitted and a received light pulse that occurs

when atmospheric particles, moving along with the wind, scatter the light pulse back to the ground based lidar (Doppler shift). Measuring this shift of frequency allows for determination of the wind speed along the line of sight (LOS), also called radial velocity, of the scanning beam. This radial velocity is linked to the orthogonal components of the three-dimensional wind vector via the following geometrical relationship:

$$v_r = u\sin(\alpha)\sin(\phi) + v\cos(\alpha)\sin(\phi) + w\cos(\phi), \tag{1}$$

where $v_r$ denotes the radial velocity along the LOS, $u$ the longitudinal wind component (east-west direction), $v$ the latitudinal wind component (north-south direction), $w$ the vertical wind component, $\alpha$ the azimuth angle and $\phi$ the zenith angle of the scanning lidar beam (90° - elevation angle). The azimuth angle is numbered clockwise starting from the north. The vertical wind speed can also be measured directly by pointing the laser beam in the vertical direction. To capture the horizontal wind components, the beams have to be tilted out of the vertical position. Under the assumption of a horizontally homogeneous wind

field across the area sampled by the scanning beams, as well as a stationary wind field for the duration of the scan, the vertical wind profile can then be inferred with suitable scanning schemes. The following sections describe the two commonly applied scanning schemes that were adopted from radar measurement technique (Lhermitte, 1969; Browning and Wexler, 1968). We replicate both of these scanning schemes in this study to derive vertical wind profiles from virtual lidar measurements.





## 2.1 VAD scanning scheme

A common scanning scheme is the velocity azimuth display method (VAD) (Kropfli, 1986; Werner, 2006). For this method, the scanning beam is tilted out of the vertical position by a fixed angle $\phi$ (zenith angle). Starting due north, the beam rotates clockwise, performing scans at different azimuth angles $\alpha$. The radial velocity is measured at each azimuth angle. In horizontally homogeneous and stationary wind conditions, plotting the radial velocities of one scan cycle against the respective azimuth angles yields a sine wave like curve. In heterogeneous and non-stationary conditions, the single measurement points scatter around the sine wave. In practice, the radial velocities are thus averaged over a certain time interval, to smooth out fluctuations. By applying a sine-wave fit to the data, one can then derive the 3D wind vector. Combining the equations for the radial velocities of one scan cycle yields a set of linear equations, which in matrix form reads:

$$A\,\boldsymbol{v} = \boldsymbol{V}_r, \tag{2}$$

where $\boldsymbol{v} = (u\,v\,w)^T$ is the three-dimensional wind vector, $\boldsymbol{V}_r = (V_{r1}, V_{r2}, V_{r3}, ... V_{rn})^T$ is a vector composed of the radial velocities measured by the respective lidar beams, and

$$A = \begin{pmatrix} \sin(\alpha_1)\sin(\phi) + \cos(\alpha_1)\sin(\phi) + \cos(\phi) \\ \sin(\alpha_2)\sin(\phi) + \cos(\alpha_2)\sin(\phi) + \cos(\phi) \\ \sin(\alpha_3)\sin(\phi) + \cos(\alpha_3)\sin(\phi) + \cos(\phi) \\ ... \\ \sin(\alpha_n)\sin(\phi) + \cos(\alpha_n)\sin(\phi) + \cos(\phi) \end{pmatrix}, \tag{3}$$

is the matrix describing the geometrical relationship between the radial velocities and the wind vector components, with n being the number of beams used for one scan cycle. The system of equations can be solved with a suitable least squares algorithm leading to the three components of the wind vector:

$$A^T A\,\boldsymbol{v} = A^T \boldsymbol{V}_r \Leftrightarrow \boldsymbol{v} = \left(A^T A\right)^{-1} A^T\,\boldsymbol{V}_r, \tag{4}$$

where $()^T$ denotes the transposed matrix and $()^{-1}$ denotes the matrix inverse. The number of beams used for one scan cycle can be varied. In this study VAD scans with 6 and 24 beams are evaluated. Due to the turbulent nature of the atmosphere, measurements are usually averaged over several scan cycles before the system of linear equations is solved to achieve approximately homogeneous conditions.

## 2.2 DBS scanning scheme

The Doppler-beam swing (DBS) technique is a simplified version of the VAD technique. In this study a total of four scanning beams is used for one complete scan cycle. The beams are tilted away from the vertical at a fixed zenith angle $\phi$ and differ in azimuth by 90°. The scan cycle starts with a beam directed towards the north, followed by the opposite (south) direction. After that the east and west direction are sampled. Each scan cycle yields the following system of equations for the three wind





components:

$$u = \frac{v_r(\alpha = 90°) - v_r(\alpha = 270°)}{2\sin\phi} \quad (5)$$

$$v = \frac{v_r(\alpha = 0°) - v_r(\alpha = 180°)}{2\sin\phi} \quad (6)$$


$$w = \frac{v_r(\alpha = 0°) + v_r(\alpha = 180°) + v_r(\alpha = 90°) + v_r(\alpha = 270°)}{4\cos\phi}, \quad (7)$$

where $u$, $v$, $w$ denote the wind components, as obtained by the lidar, $v_r$ are the radial velocities measured at the respective azimuth angle $\alpha$, and $\phi$ is the zenith angle of the lidar. Because the DBS scanning scheme requires only four beams, one scan cycle is completed faster than for the VAD scheme using 6 or even 24 beams. This means that for the same time interval, more

scans are available for averaging.

The virtual lidar simulator used in this study replicates both scanning schemes (VAD and DBS) simultaneously, which allows for a direct comparison.

## 3 Simulation of lidar measurements

Because we aim to investigate the lidar measurement error that occurs due to inhomogeneous and turbulent flow structures,

it is essential to use a simulation model that represents turbulent atmospheric motions on a scale similar to the resolution of the lidar. Large-Eddy Simulation (LES) technique explicitly resolves the bulk of turbulent motion and only parameterizes the remaining part. It is therefore a suitable technique for this study, provided that the spatial resolution of the model is sufficient. The wind fields, as well as the integrated virtual lidar measurements are simulated using PALM (PArallelized Large-eddy simulation Model) (Raasch and Schröter, 2001; Maronga et al., 2015). By default, PALM applies the spatially filtered

Boussinesq-approximated form of the Navier-Stokes equations, treating the flow as incompressible but allowing for variations in density due to buoyancy. The model solves the governing prognostic equations on a Cartesian grid using finite differences. For improved resolution, a staggered grid (Arakawa and Lamb, 1977) is employed. Implicit filtering of the flow variables is realized through spatial discretization. Effects of the filtered small-scale turbulence are represented by a sub-grid-scale parameterization according to Deardorff (1980). For this study we use PALM in version 6.0, revision 4856.


### 3.1 Virtual lidar

We implemented virtual lidar measurements in the user module of PALM, which is an interface that allows users to define custom output variables. The lidar simulation tool was able to simulate multiple lidars with different zenith angle configurations simultaneously. This allowed for direct comparison of measurements obtained with different scanning configurations.





We designed the virtual lidar measurements such that three lidars are placed in the center of the model domain. The zenith angle $\phi$ of the tilted scan beams is different for each lidar ($\phi = \in \{15°, 30°, 54.7°\}$). Different scanning schemes (VAD and DBS) are performed simultaneously by each of the virtual lidars.

       For the VAD method with 24 beams, the process of the virtual measurement is as follows: The modeled flow field is probed along the LOS of the tilted lidar beams. During a full scan cycle the beam traces a conical shape. The atmosphere is scanned
in azimuth direction at equidistant steps of 15°, starting north ($\alpha = 0°$). To imitate the time a real instrument needs to send and receive a signal and change the position of its scanning beam, the time for 'measuring' the radial velocity in one direction is set to five seconds. Hence, a complete scan with 24 beams takes 120 seconds. A scan configuration using less than 24 beams completes multiple revolutions during the same time interval. For example, a scan with six beams completes four revolutions in 120 seconds. In that case, the measured radial velocities are averaged over these four revolutions before being processed
further.

       The radial velocity along the beam is derived for every height (grid) level according to Eq. 1. Instead of interpolating, the velocity components $u$, $v$ and $w$ at the grid point closest to the beam are used. Due to the staggered grid employed in PALM, $u$, $v$ and $w$ from different grid points are included in the calculation. The radial velocities for the respective directions are stored in a netCDF file, which is further processed after the simulation.

The velocity profiles as 'measured' by the virtual lidar are calculated from the radial velocities along the scanning beams. First, the radial velocities are averaged over the desired averaging period. Second, the linear system of equations described by Eq. 2 is solved via a least squares algorithm using the function linalg.lstsq of the numerical python package (Harris et al., 2020). This yields the three wind components $u$, $v$ and $w$ as measured by the lidar. These lidar profiles are later compared to the 'true' profiles as provided by the LES. For the DBS method, lidar profiles are calculated using Eq. 5 - Eq. 7.

The initial scan direction (azimuth angle) can be changed to a different value than 0°, thereby rotating the virtual instrument. This allows for an investigation of a possible effect of the instrument's orientation relative to the main flow direction on the accuracy on the measurement.

       Lidar range-gate-length averaging effects are not a-priori considered by the virtual lidar simulator. Since we are essentially interested in the errors resulting for the derived wind vector at a given height or as an average across the entire ABL due to the
scanning strategy, these effects will not be discussed in this study.

## 3.2    Quantification of lidar error

For quantification of the lidar retrieval error we use the root-mean-square deviation (RMSD) as a bulk measure for the cumulative error within the entire boundary layer. It is defined as the square root of the quadratic mean difference between the





predicted values (in this case, the true values at the respective grid point) and the observed value (the values measured by the
lidar):

$$
\text{RMSD} = \sqrt{\left(\frac{\sum\limits_{i}^{n}\left(V_t - V_m\right)}{(n-i)}\right)^2},
\tag{8}
$$

where $V_t$ is the true wind speed (or wind direction) as generated by the LES at the respected height level and $V_m$ is the
wind speed (or wind direction) measured by the lidar. In this case the mean refers to the mean over the height interval, which
is enclosed by a lower boundary $i$ and an upper boundary $n$. A lower RMSD value implies a better agreement between the
data than a higher one, while a zero RMSD indicates perfect agreement. The following procedure was applied to each simu-
lation case: During the simulation virtual lidar measurements were performed. Afterwards, the measurements (as well as the
reference profiles) were averaged over a chosen time interval and the wind vector was derived according to the selected scan
scheme (Sec. 2). The resulting profiles of horizontal wind speed and wind direction were compared to the truth values of the
simulation along the vertical column above the lidar location. The difference between the value 'measured' by the lidar and the
truth value was calculated at each grid point along this vertical column. Subsequently, RMSD was calculated over the vertical
extent of the boundary layer, which was estimated for each simulation case individually.

Note that measurement of the horizontal wind with lidar scan technique requires the lidar beams to be tilted out of the verti-
cal. This results in a conical scan above the lidar location, with the scan cone widening with increasing height. This implies that
the lidar does not exactly probe the vertical column above its location. Instead, values measured by the instrument represent
the velocity of the air within the circle spanned by the cone at the respective height. As it is usually the vertical wind profile
at a certain location that is of interest for NWP models, we chose to compare the lidar measurements to the LES values at the
respective grid points directly above the lidar.

## 4   Simulation setups

We performed measurements with the virtual lidar during different simulation runs. We chose four cases, for which we applied
varying geostrophic forcing and surface heating to generate different types of flow regimes from buoyancy to shear driven
flows. Depending on the intensity of buoyancy and shear, the flow will develop characteristic features that take shape in the
form of elongated streaks, rolls or cells. These coherent structures are typical for the turbulent ABL and are well described in
literature (Deardorff, 1972; Moeng and Sullivan, 1994; Khanna and Brasseur, 1998; Salesky et al., 2017). They determine the
flow morphology and are hence expected to influence the measurements of wind profiles with lidar scanning schemes.

We chose a moderately convective situation as the basis case (case 1). The simulation setup for this case is described below.
The simulation setups for the other cases were identical, differing only in prescribed geostrophic wind and, in one case, surface
heat flux. Table 1 summarizes the simulation properties.





**Table 1.** Properties of the simulation cases: geostrophic wind speed ($u_\mathrm{g}$), sensible heat flux at the surface ($H_0$), boundary layer height ($z_i$), Obukhov length ($L$) and friction velocity ($u_*$).

| Case number | $u_\mathrm{g}$ | $H_0$ | $z_i$ | $L$ | $u_*$ |
|---|---|---|---|---|---|
| | $\mathrm{m\,s^{-1}}$ | $\mathrm{W\,m^{-2}}$ | m | m | $\mathrm{m\,s^{-1}}$ |
| 1 | 5 | 150 | 960 | -20 | 0.30 |
| 2 | 10 | 75 | 860 | -140 | 0.47 |
| 3 | 2 | 150 | 960 | -5 | 0.12 |
| 4 | 0 | 150 | 960 | -3 | 0.15 |

For all simulations we chose a domain size of $4 \times 4 \times 2\,\mathrm{km}^3$, to allow for the development of characteristic turbulent
structures and to ensure a sufficiently large horizontal extension of the domain for the lidar scan cone to fit even at large zenith
angles and higher levels. Since LES are able to resolve eddies that are about 5 - 6 times as large as the grid spacing (Chow and
Moin, 2003; Cheinet and Siebesma, 2009) and the lidar resolution range lies in the order of a few tens of meters, a grid size of
$\Delta = 5\,\mathrm{m}$ will be sufficiently small to resolve turbulent motions that are relevant for the lidar measurement. At the beginning
of the simulation, all atmospheric variables were prescribed by vertical profiles, assuming horizontal homogeneity. Boundary
conditions were set to be cyclic in all lateral directions, which implies a periodic continuation of the flow field. Flow structures
leaving the border of the domain re-enter at the opposite boundary. The initial temperature profile was specified as neutrally
stratified up to a height of $800\,\mathrm{m}$ in order to speed up the development of a convective layer of the desired height. Above $800\,\mathrm{m}$,
the boundary layer was topped with an inversion with a potential temperature gradient of $1\,\mathrm{K}/100\,\mathrm{m}$. At the surface layer a
sensible heat flux of approximately $150\,\mathrm{W\,m^{-2}}$ was prescribed. The flow was driven by a large-scale pressure gradient which
corresponds to a geostrophic wind of $5\,\mathrm{m\,s^{-1}}$. The flow entered the simulation domain from the west, so that $\mathrm{v}_\mathrm{g} = 0\,\mathrm{m\,s^{-1}}$.
To keep the boundary layer height constant for the duration of the simulation, a large-scale subsidence of $-0.018\,\mathrm{m\,s^{-1}}$ was
applied at $1000\,\mathrm{m}$ and above, which decreased linearly to zero between $1000\,\mathrm{m}$ and $0\,\mathrm{m}$. The surface was homogeneous with a
roughness length of $0.05\,\mathrm{m}$. Random perturbations with small amplitudes were imposed onto the horizontal wind field during
the initial stage of the simulation to trigger the onset of turbulence. The simulated time amounted to a total of 3 hours, whereby
the first 2 hours are to be considered as spin up time. After this time the model reached a quasi-stationary turbulent state. The
analysis of data extended over 1 hour, starting after spin up time. A constant time step of 0.2 seconds was used, in order to allow
for a uniform data output at a time interval of 5 seconds. A constant time step was necessary to ensure the correct functionality
of the virtual lidar simulator.



## 4.1 Ensemble simulations

A single simulation of lidar measurements in the turbulent atmosphere will help to get a rough estimate of the error associated with the violation of the homogeneity assumption. However, a generalization of this estimate, derived from a single manifestation of a turbulent flow is most likely deficient. To characterize the statistical uncertainty of the measurement error, we chose two simulation cases (case 1 and case 4) for which we performed an ensemble of simulations. For each of the two cases we carried out ten member simulations with slightly different initial conditions, which lead to to the development of different specific realizations of the turbulent flow fields. Averaging the lidar error over all ten realizations approximates the ensemble average (Wyngaard, 2010), which provides a statistical measure for the error to be expected from lidar wind profiling. The size of ten members for each ensemble was chosen with regard to limited computational resources, constituting a compromise between the theoretical necessity of a larger ensemble and the practical feasibility.

## 5 Results

### 5.1 Lidar profiles in a moderately convective boundary layer (basis case 1)

In general, an LES needs a certain period of time until a turbulent flow fully develops, and the simulation reaches a quasi-stationary state. After this spin up time has elapsed, the analysis of virtual lidar measurements can start. To determine if turbulence has fully developed and if the simulation has reached a quasi-stationary state, it is useful to examine the time series of turbulence parameters. Figure 1 shows time series data of domain averaged resolved-scale turbulence kinetic energy ($E^*$), friction velocity ($u^*$) and maximum velocity of the u-component ($u_{max}$) for an exemplary member simulation of case 1. After a simulation time of 2 hours all quantities reached a quasi-stationary state. $E^*$ stabilized approximately at a value of $0.45\,\mathrm{m^2\,s^{-2}}$, $u^*$ at approximately $0.30\,\mathrm{m\,s^{-1}}$ and $u_{max}$ at approximately $8.75\,\mathrm{m\,s^{-1}}$. Analysis of virtual lidar measurements started after two hours of simulated time.

The flow structures occurring in various boundary layer regimes have been investigated with LES and described in many studies (e.g. Deardorff (1972); Moeng and Sullivan (1994); Khanna and Brasseur (1998); Salesky et al. (2017)). As expected, the instantaneous flow field at a simulation time of 2 hours (Fig. 2) exhibits organizational patterns typical for a moderately convective flow regime, where buoyancy forces interact with shear. The vertical velocity field exhibits elongated cellular structures. Near the surface these structures are comparatively fine and orientated along the mean wind direction, with narrow regions of updraft, surrounded by broader areas of slightly weaker downdraft (Deardorff, 1972). This pattern gets wider at greater heights, the lines turn into large patches, while the velocities increase. The fields of the horizontal wind components display a more patchy pattern, with alternating regions of high- and low-speed flow, which suggests the formation of horizontal convection rolls (Moeng and Sullivan, 1994; Khanna and Brasseur, 1998; Salesky et al., 2017). The orientation of the structures deviates from the mean wind direction. At greater heights the patches fade out and velocity differences decrease.

These coherent structures which are present in the turbulent flow field indicate that the requirement of a horizontally homogeneous wind field within the air volume scanned by the lidar is not met by the instantaneous flow field. As the scanning





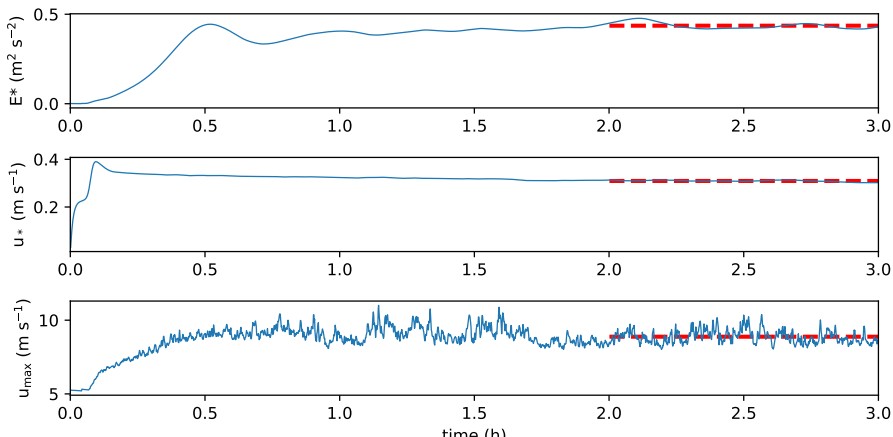

**Figure 1.** Simulation case 1: Time series of resolved-scale turbulence kinetic energy (E*), friction velocity (u*) and maximum velocity of the u-component ($u_{max}$). All quantities are averaged over the entire 3D-domain. The dashed red lines indicate the respective mean value between 2 h and 3 h of simulated time.

cone of the lidar expands with increasing height, it encloses areas of different velocities (Fig 2 upper panels). It therefore becomes clear that temporal averaging is necessary for a successful wind retrieval from the lidar scan under convective conditions.

### 5.1.1 Effect of averaging time and zenith angle configuration

Since coherent structures are present in the instantaneous flow field, the prerequisite of horizontal homogeneity, required for wind vector retrieval with lidar scanning schemes is not fulfilled. Temporal averaging will lead to a more homogeneous flow field, because the structures responsible for inhomogeneity are moving along with the mean wind and are thus generally averaged out over time. Averaging the radial measurements over a certain time period, before retrieving the wind profile, we expect to improve the accuracy of lidar measurements. However, it is not clear how long the averaging interval must be, so that the assumption of horizontal homogeneity applies. Therefore, we first examine the behavior of lidar retrieval error for different averaging periods. For this part of the analysis, we restrict the scanning scheme to the VAD method with 24 beams (VAD 24).

Figure 3 shows vertical wind profiles measured with different zenith angle configurations for averaging intervals of 10, 30 and 60 min, respectively. It becomes apparent that deviations from reference can vary significantly with height, but become minimal near the top of the boundary layer. Deviations of lidar profiles from the true profile mostly decrease with increasing averaging time. This is independent of the zenith angle of the scanning lidar and applies to wind speed, as well as to wind direction and vertical wind. However, exceptions can occur. For example, for the 54.7° zenith angle configuration (yellow line) the 30-min average (bottom, center) yields larger deviations from reference that the 10-min average (bottom left).

We quantify the results for this exemplary analysis via the RMSD (Fig. 4), which is calculated over a height interval extending from 50 m to 960 m. Note that all values are mean values over a 60-min measurement period. Thus, there is one singular





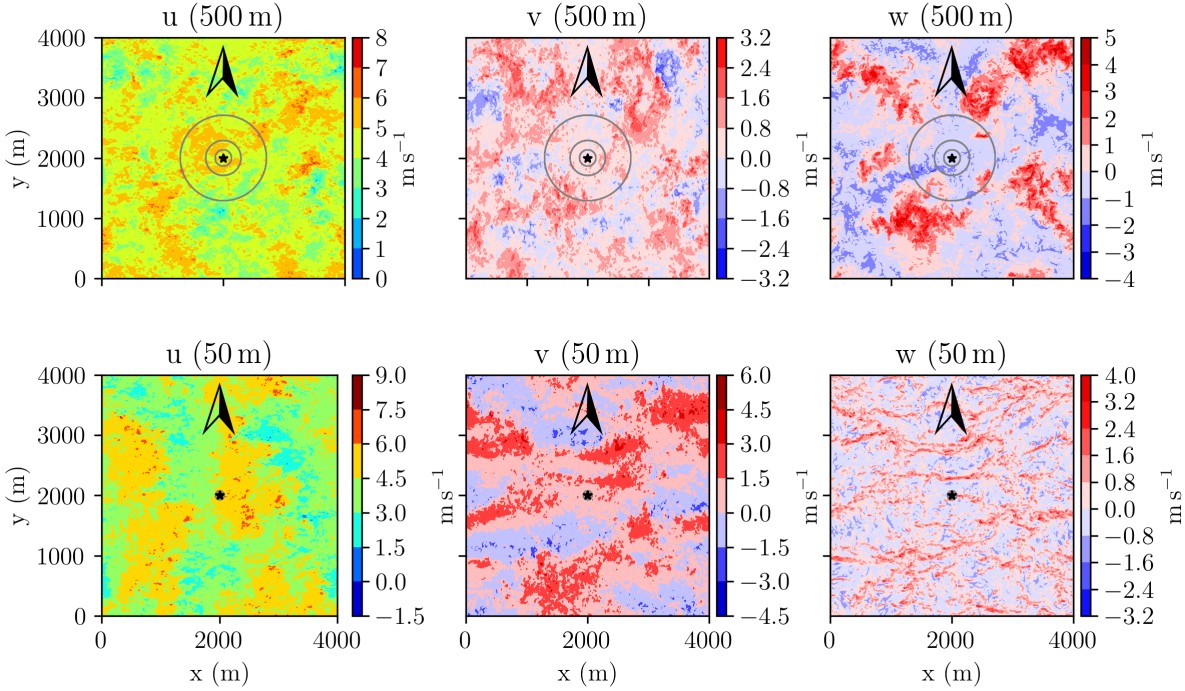

**Figure 2.** Simulation case 1: Horizontal cross sections of instantaneous flow field components (u, v, w) at 2 h simulation time, at 50 m and at 500 m height. The black star marks the location of the three virtual lidars, placed in the center of the domain. Each of the lidars is configured with a different zenith angle (15°, 30°, 54.7°). Gray circles indicate the corresponding scanning circles at the respective height level. Arrows are pointing north.

value for the 60-min interval, while for the 30 (10) -min interval, the value is an average of two (six) values. The results confirm that the lidar retrieval error decreases with increasing averaging time. The RMSD for wind speed ranges from $0.18 \, \mathrm{m \, s^{-1}}$ to approximately $0.55 \, \mathrm{m \, s^{-1}}$ for a 10-min averaging period, depending on the zenith angle configuration. Extending the averaging time to 60 min results in RMSD below $0.2 \, \mathrm{m \, s^{-1}}$. Errors for wind direction range from 4° to 13° for the 10-min average, while reducing to less than 4° for the 60-min average. RMSD for the vertical wind component range from $0.1 \, \mathrm{m \, s^{-1}}$ to about $0.3 \, \mathrm{m \, s^{-1}}$, while remaining below $0.1 \, \mathrm{m \, s^{-1}}$ for measurements averaged over 60 min.

Comparing different zenith angle configurations, we observe that the lidar error for wind speed measurements decreases with increasing zenith angle. The differences between the configurations become smaller for larger averaging intervals, while for a 10-min averaging period the differences are substantial. Here, the RMSD gap between the smallest zenith angle and the largest angle can be three times as large as the smallest RMSD, indicating that the choice of zenith angle is critical for shorter averaging periods. An opposite trend can be observed for the vertical wind. Here, it is the largest zenith angle configuration that yields the largest lidar errors. For the wind direction, no clear dependency can be detected.

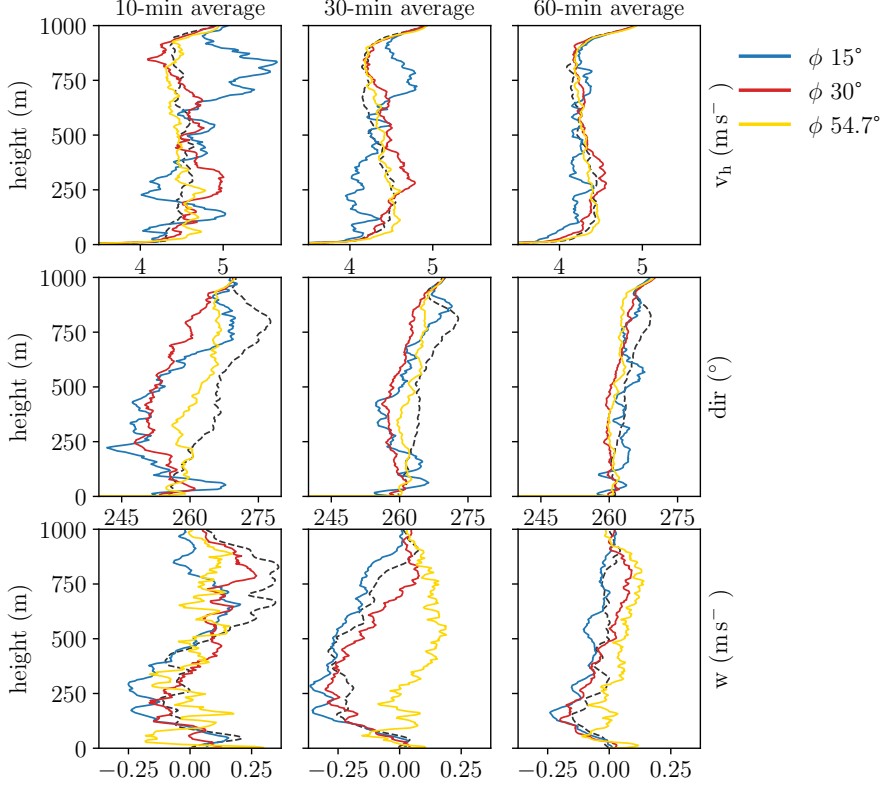

**Figure 3.** Simulation case 1: Time averaged vertical wind profiles of lidar scans with the VAD 24 technique using different zenith angle configurations ($\phi$). Displayed are horizontal wind speed (upper panels), wind direction (center panels) and vertical wind (lower panels). Different colors indicate different zenith angle configurations. Reference profiles (true LES values) along the vertical column directly above the lidar location are shown in black (dashed lines).

### 5.1.2 Effect of scanning scheme

Subsequently, we investigate if and how using a different number of scan beams as well as changing the orientation of the lidar with respect to the flow direction affects the accuracy of lidar measurements. Simultaneously, we evaluate different scanning schemes regarding their performance. For this purpose, we compare virtual measurements conducted with a fixed zenith angle of 15° but with different scanning schemes. We investigate a VAD scheme with six beams (VAD 6), a VAD scheme with 24 beams (VAD 24) and a DBS scheme. By default, both scanning schemes, VAD as well as DBS, start at an azimuth direction of 0° north. For each configuration, two virtual measurements were performed simultaneously, with one of the lidars rotated. For the VAD scans, the respective second lidar was rotated 30° to the east, which constitutes half of the azimuth step used for the VAD 6 scheme. For the DBS scan, the second lidar was rotated 45° to the east, which also constitutes half of the azimuth step used for this scheme. We again considered averaging intervals of 10, 30 and 60 min, but decided to display profiles only





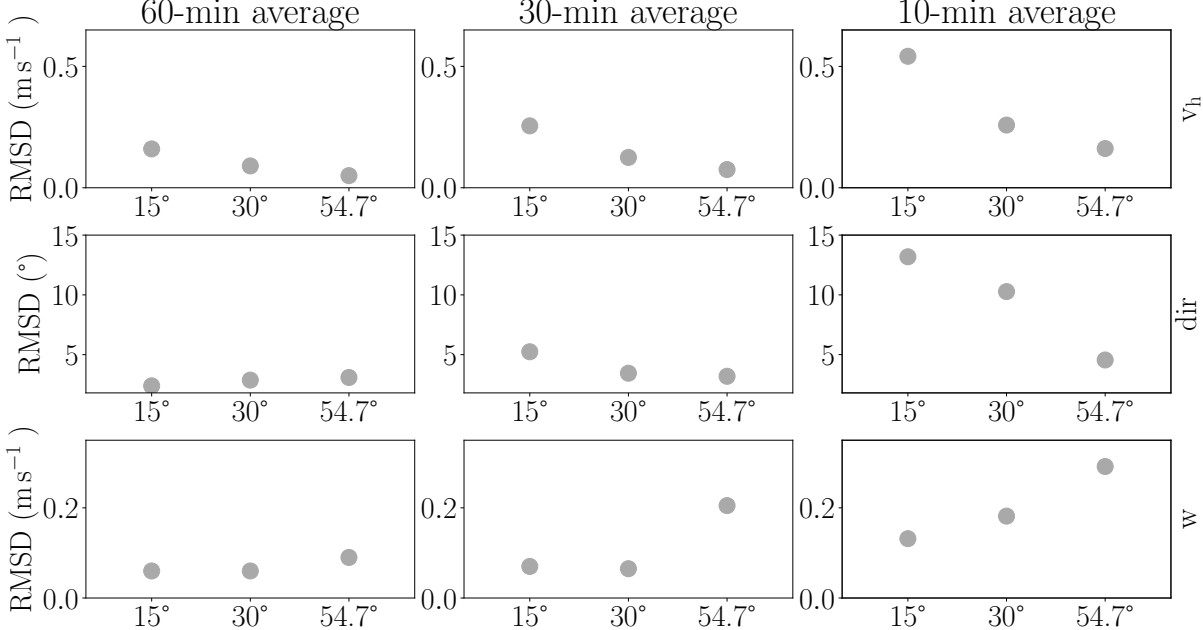

**Figure 4.** Simulation case 1: RMSD of virtual lidar measurements of wind speed, wind direction and vertical wind with reference to the 'true' wind profile of the column located directly above the lidar. RMSD values were calculated over a height interval extending from 50 to 960 m for measurements averaged over different time intervals. Shown are results for measurements obtained with the VAD 24 scan technique at different zenith angles (15°, 30° and 54.7°).

for the 10-min average, for reasons of clarity.

Profiles retrieved with different scanning schemes clearly deviate from each other and rotated lidars yield slightly different
measurement results than their north-oriented counter-parts (Fig. 5). However, differences between rotated lidars are rather small. Respective RMSD are mostly close to identical (Fig. 6) and differences are limited to less than $0.01\,\mathrm{m\,s^{-1}}$ for wind speed, $2°$ for wind direction and $0.05\,\mathrm{m\,s^{-1}}$ for vertical wind. There is no definite answer as to which orientation is more advantageous. In some instances, it is the north-oriented lidar that measures more accurately, while in other instances it is the rotated lidar that exhibits the smaller error.

The VAD 24 configured lidar yields the least accurate results for measurements of wind speed, whereas instruments using the VAD 6 or the DBS technique perform more accurately (Fig. 6). The discrepancies between the configurations reduces with increasing averaging periods. While the difference amounts to approximately $0.25\,\mathrm{m\,s^{-1}}$ for the 10-min average, it diminishes to only $0.1\,\mathrm{m\,s^{-1}}$ for the 60-min average. For the wind direction, as well as the vertical wind, no configuration stands out as particularly advantageous.


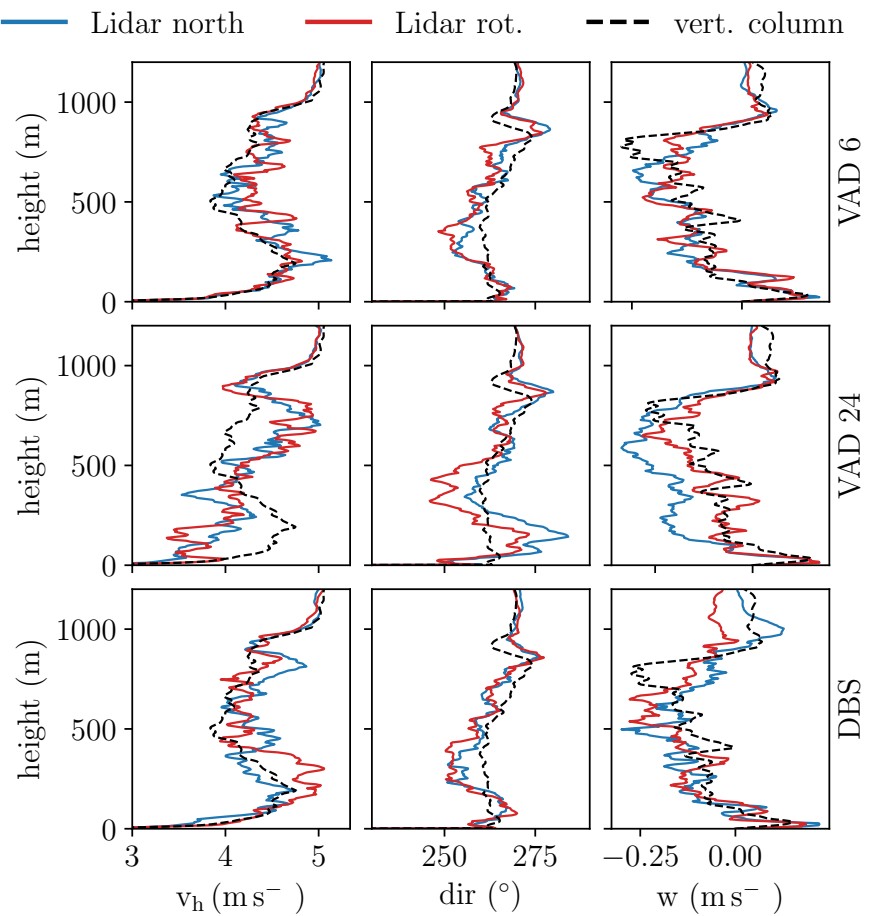

**Figure 5.** Simulation Case 1: Virtual lidar profiles obtained over a 10-min period with a zenith angle configuration of 15° using different scan schemes. Blue lines indicate profiles of north-oriented lidars and red lines indicate profiles of rotated lidars. For the VAD scans, lidars were rotated 30° to the east, while for the DBS scan, the lidar was rotated 45° to the east. 'Truth' profiles (LES values) along the vertical column directly above the lidar location (black dashed lines) are shown for reference.

## 5.2 Lidar profiles in different boundary layer regimes

A variation of geostrophic wind and surface heat flux generates different boundary layer regimes. Increasing the geostrophic wind enforces vertical shear while increasing the surface heating strengthens buoyancy driven thermals. If buoyancy and shear are both present in the ABL, they interact to modify the flow structure. In a more shear driven regime, turbulent eddies gener-

ally organize into elongated bands or streaks, while buoyancy driven flows tend to form concentrated areas of strong updraft surrounded by broader areas of downdraft (Moeng and Sullivan, 1994). Since different flow regimes produce flow structures of different size and shape, we can expect that the flow regime has a non-negligible effect on the lidar measurements. To



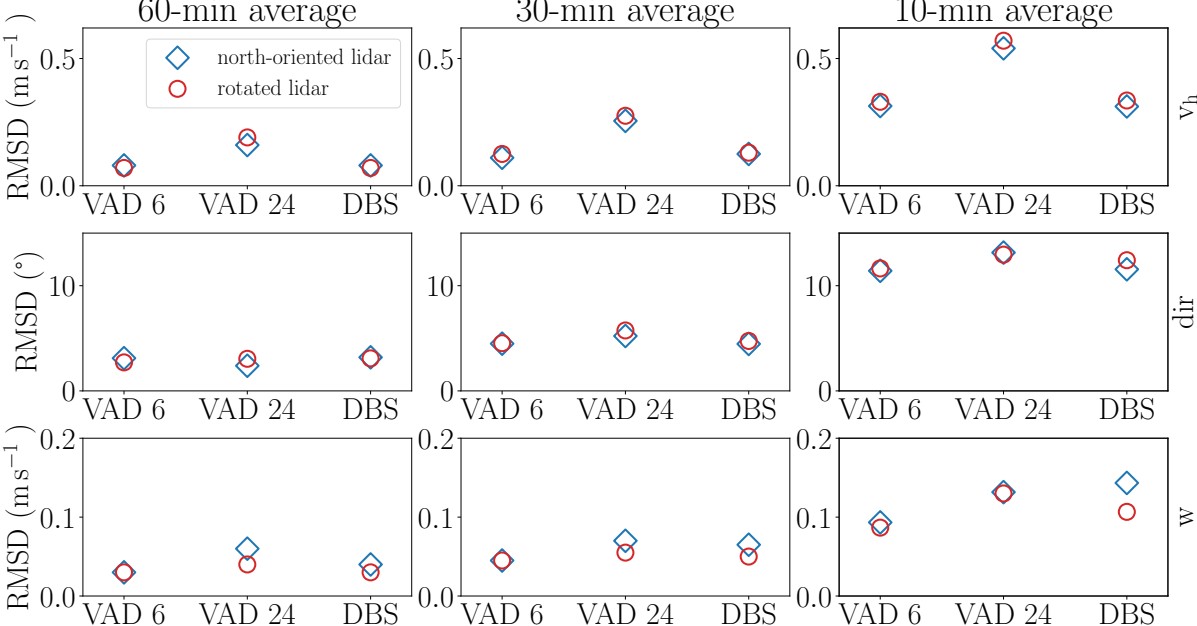

**Figure 6.** Simulation case 1: RMSD of virtual lidar measurements of wind speed, wind direction and vertical wind with reference to the 'true' wind profile of the column located directly above the lidar. RMSD values were calculated over a height interval extending from 50 to 960 m for measurements averaged over different time intervals. Shown are results for measurements with a zenith angle configuration of 15°. Measurements were obtained with different scan schemes (VAD 6, VAD 24 and DBS). Blue diamonds indicate results for north-oriented instruments, while red circles mark results for rotated instruments. Lidars were rotated 30° to the east for VAD scans and 45° to the east for DBS scans.

study how the lidar error depends on the flow regime, we simulated two additional convective boundary layers with different geostrophic forcing and sensible surface heat flux (Table 1). For cases 2 and 3 we focus on different scanning schemes and thus

restrict our discussion of the virtual lidar measurements to a zenith angle configuration of 15°.

For case 2 (case 3) RMSD was calculated over a height interval from 50 to 860 m (from 50 to 960 m) representing the top of the convective ABL in these cases, respectively. Note that the absolute values were not normalized with regard to the averaged wind speed of the respective simulation. This should be kept in mind when comparing the values, as the background winds of the two simulations differ considerably. However, non-normalized values have the advantage of being more straightforward,

and providing direct information on the error made by the lidar, which is most relevant for practical application.

RMSD values are generally smaller for case 2 than for case 3 (Fig. 7), suggesting a higher accuracy of lidar measurements for higher wind speeds. The reduced wind speed in case 3 favors the generation of more buoyancy driven structures, while buoyancy induced turbulent structures are damped in case 2 by the reduced surface heat flux and the increased wind speed, leading to a more horizontally uniform uniform distribution of the wind field. This suggests, that we find more optimal con-

ditions for wind vector retrieval with lidar measurement schemes under shear driven conditions rather than under buoyancy



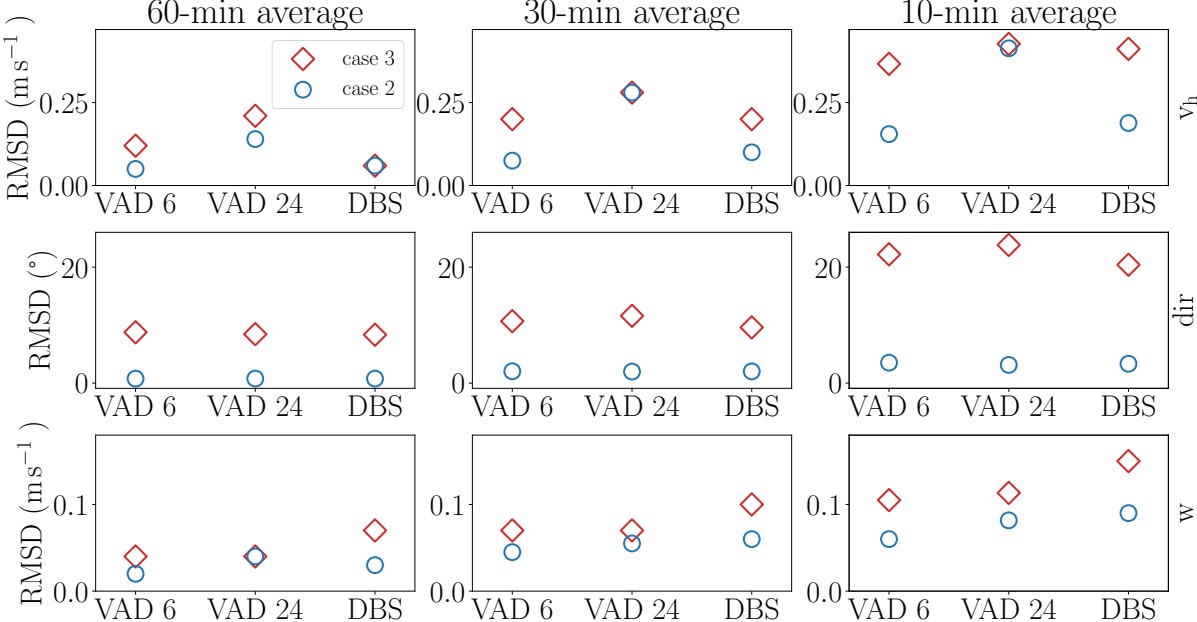

**Figure 7.** Simulation cases 2 and 3: RMSD of virtual lidar measurements of wind speed, wind direction and vertical wind with reference to the 'true' wind profile of the column located directly above the lidar. RMSD values were calculated over a height interval extending from 50 to 860 m (from 50 to 960 m) for case 2 (case 3). Measurements are averaged over different time intervals. Shown are results for measurements with a zenith angle configuration of 15°. Measurements were obtained with different scan schemes (VAD 6, VAD 24 and DBS). Blue circles denote results for simulation case 2, while red diamonds indicate results for simulation case 3.

driven conditions. In some instances, however, error values for the two cases are remarkably close (Fig. 7 top center panel) or even the same (Fig. 7 top left panel and bottom left panel).

In both cases RMSD for wind speed are larger again for VAD 24 configurations when compared to VAD 6 configurations. This feature, which had already been observed in case 1, is more pronounced for case 2 than for case 3 and is most noticeable in the 10-min average. Noticeably large RMSD occur in case 3 for the wind direction, with maximal values exceeding 20 degrees. This can be attributed to the buoyancy induced motions in this case, which lead to a rather pronounced fluctuation of the wind field. This entails a less precise determination of the mean wind direction. RMSD values for the vertical component of both simulation cases, however, are comparable to case 1.

We would like to mention that in addition to the convective cases discussed here, we considered the case of a stably stratified boundary layer as well. Due to the comparably small flow structures, the lidar measurement errors were very small and the choice of scan strategy did not affect the results significantly. We therefore decided to refrain from discussing the results from the stable case.

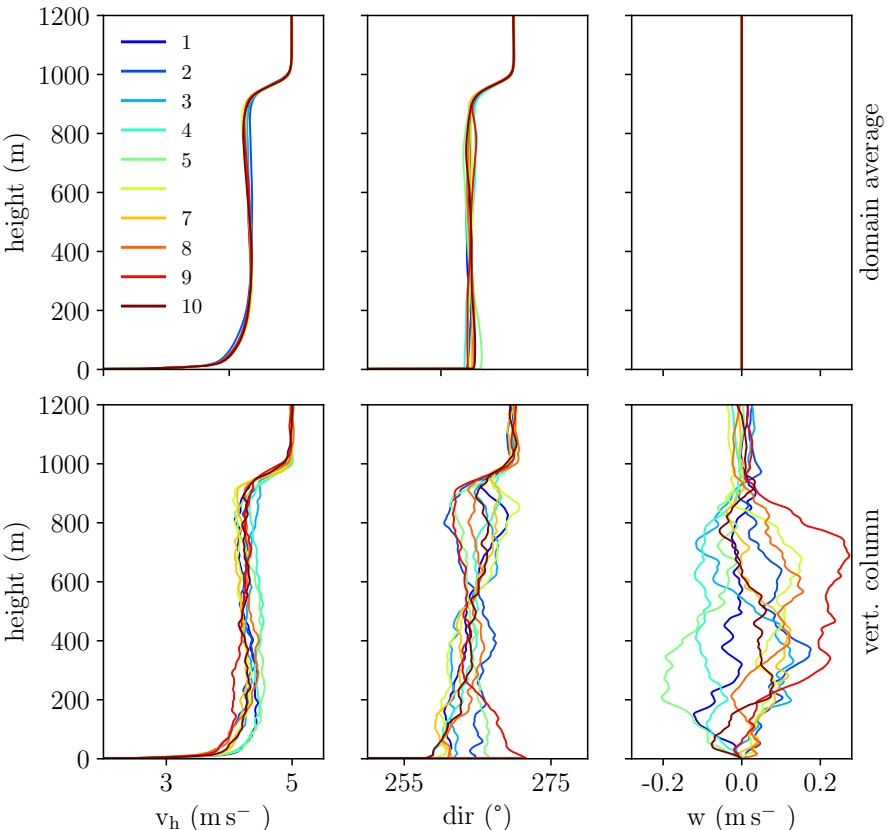

**Figure 8.** Ensemble case 1: Vertical profiles of wind speed, wind direction and vertical wind of every member of the ensemble simulation averaged over a period of 60 min. Upper panels show horizontally averaged profiles, while lower panels display profiles of the vertical column directly above the lidar location. Straight lines at zero value for the domain average of the vertical wind indicate incompressibility of the flow field.

### 5.3 Ensemble simulations

To investigate how reliable the results found for a single simulation are, we extend the individual simulation of case 1 to an
ensemble of 10 members. For each member simulation the same setup is used, but the random perturbations, that are imposed onto the flow field at the beginning of the simulation, are varied, which leads to the development of unique realizations of the turbulent flow field. Profiles of wind speed, wind direction and vertical wind, averaged horizontally over the simulation domain and temporally over one hour, vary only marginally between members (Fig. 8 top row), indicating that the simulation runs share the same statistical properties, whereas profiles of the vertical columns above the lidar location (Fig. 8 bottom row) show
greater variability, illustrating the unique turbulent flow realization of each member.

Figure 9 shows a statistical evaluation of all virtual measurements of the ensemble. The evaluation confirms that the error generally decreases with increasing averaging time. Not only the ensemble mean error decreases, but the range from smallest





to largest occurring error contracts as well, indicating that the probability of large errors considerably decreases. Note that unlike Fig. 4, where only one single simulation was evaluated, the values displayed here are statistical ones. In this case, the

statistical population comprises the ten members of the ensemble, meaning that for the 60-min averaging interval, ten values enter the statistics. For the 30-min averaging interval 20 values are available, while for the 10-min interval 60 values enter the statistical analysis.

For wind speed measurements, RMSD values are less than $1\,\mathrm{m\,s^{-1}}$ for all configurations and members, with mean values below $0.6\,\mathrm{m\,s^{-1}}$. Most accurate measurements are obtained with a zenith angle of 54.7°, which, again, confirms the results

found for basis case 1. Outstanding results are accomplished with a combination of a 54.7° zenith angle with the DBS scan scheme. This configuration achieves the most accurate measurements and exhibits the smallest range of error. The least accurate results, conversely, are delivered by the VAD 24 scans with a zenith angle of 15°. Again, the errors of VAD 6 are smaller than those of VAD 24 for most of the cases considered.

RMSD values for wind direction range from 1° to 23°, with mean values below 10°. A larger zenith angle proves slightly

more favorable for measurements of wind direction, but no scanning scheme stands out.

Mean error values for measurements of the vertical wind component are less than $0.6\,\mathrm{m\,s^{-1}}$. Lowest error values occur with the smallest zenith angle (15°) in conjunction with a VAD scanning regime. The largest errors are obtained with a large zenith angle (54.7°) in conjunction with the DBS technique (maximum RMSD greater than $1.6\,\mathrm{m\,s^{-1}}$ for an averaging period of 10 min).

Using the same approach, we analyze a second ensemble of simulations for simulation case 4, which features a purely convective situation without any geostrophic background wind. This is a simplified and idealized case that allows to investigate the limits of lidar wind profiling under extreme conditions.

It should be noted that, because there is no geostrophic forcing applied in case 4, the only source of turbulent motion is the sensible heat flux at the surface. In such a purely convective situation, where no mean background wind is present, turbulent

structures do not travel across the domain but are rather stationary during their life cycle. The lidar error is thus expected to be larger, because the assumption of homogeneity is violated even for longer averaging times. Also, since there is no background wind, there is little meaning in measuring a mean wind direction. Therefore, in this case, we refrain from discussing the wind direction.

The statistical evaluation of all virtual measurements of the ensemble (Fig. 10), exhibits general similarities to ensemble case

1. Like in ensemble case 1, lidar errors for wind speed and vertical wind decrease with increasing averaging time. However, mean deviations from truth values are significantly larger than in case 1. RMSD for wind speed reaches a maximum of $3\,\mathrm{m\,s^{-1}}$, with mean values ranging from $0.3\,\mathrm{m\,s^{-1}}$ to $1.3\,\mathrm{m\,s^{-1}}$, depending on scan configuration and averaging period. We confirm that mean deviation for wind speed is generally smallest for measurements with a zenith angle of 54.7°. The differences in error behavior between the three zenith angle configurations are even more pronounced in the purely convective boundary layer than

in the moderately convective case. In contrast to ensemble case 1, a combination of VAD 24 with a zenith angle of 15° does not produce the largest errors in ensemble case 4. Instead, the DBS scheme exhibits the largest deviations. Of all convective simulation cases analyzed, this is the only case for which a configuration of VAD 6 in conjunction with a small zenith angle is





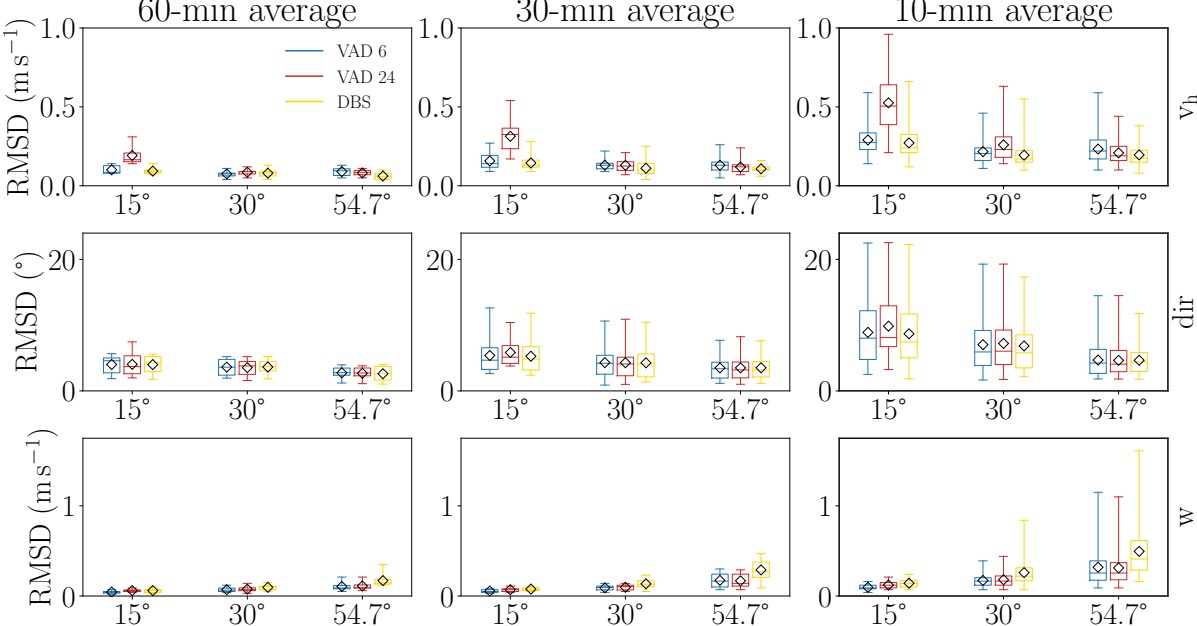

**Figure 9.** Ensemble case 1: RMSD of virtual lidar measurements of wind speed, wind direction and vertical wind with reference to the respective 'true' wind profile above the lidar. RMSD values were calculated over a height interval extending from $50\,\mathrm{m}$ to $960\,\mathrm{m}$ for measurements averaged over different time intervals. Results are grouped by zenith angle configuration (15°, 30° and 54.7°). Different colors denote different scan techniques (VAD 6, VAD 24 and DBS). Whiskers indicate the entire range of values, including the maximum and minimum. In addition, mean values are marked by a black diamond.

not superior to VAD 24 with a small zenith angle but it is also not worse.

Similar to ensemble case 1, measurements with the largest zenith angle generally yield the largest deviations for the vertical wind component. Mean RMSD for the vertical wind are below $1\,\mathrm{m\,s^{-1}}$. Maximal errors, however, exceed $2\,\mathrm{m\,s^{-1}}$. The best performance is achieved with the VAD scan technique in conjunction with a 15° zenith angle configuration. In contrast, the least accurate measurements are obtained with the largest zenith angle, using the DBS technique.

## 6 Discussion and conclusions

This study uses virtual measurements implemented in LES to investigate different scan strategies for wind profiling with Doppler-lidar instruments over a homogeneous and flat surface. Comparing the virtual measurements with the truth value of the LES, we are able to quantify the retrieval error that arises due to the violation of the homogeneous wind field assumption. Various factors influencing this error, such as averaging time, zenith angle configuration, scanning scheme and instrument orientation are analyzed for different meteorological situations. By extending two selected simulation cases to an ensemble each,

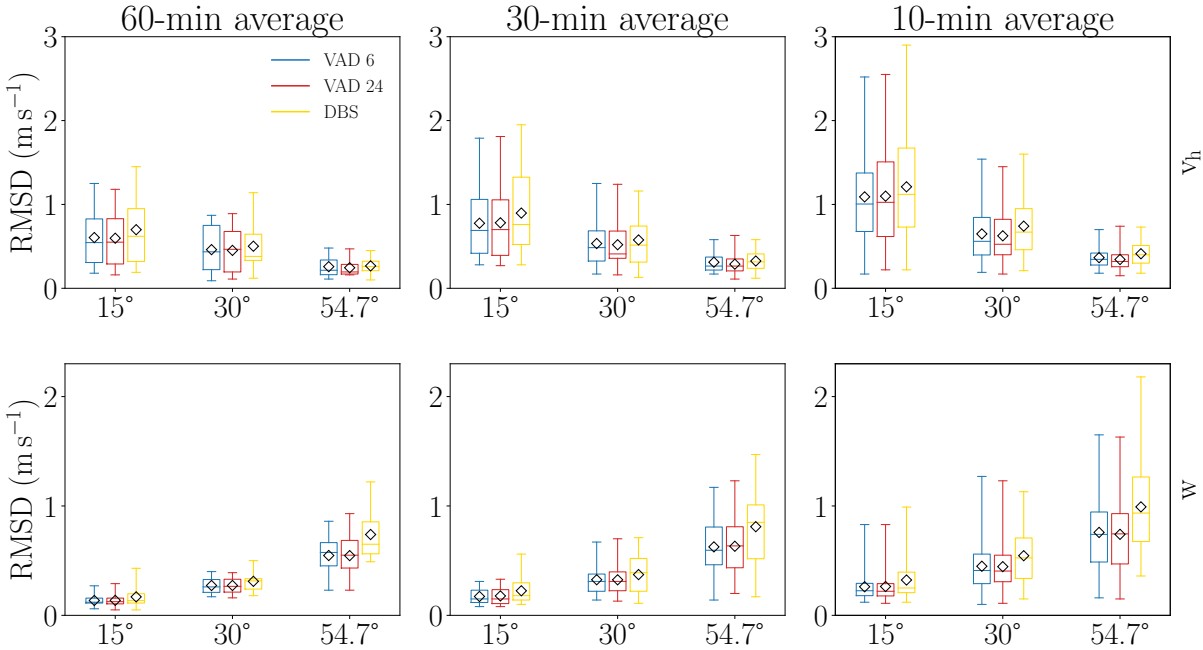

**Figure 10.** Ensemble case 4: RMSD of virtual lidar measurements of horizontal and vertical wind with reference to the respective 'true' wind profile above the lidar. RMSD values were calculated over a height interval extending from 50 m to 960 m for measurements averaged over different time intervals. Results are grouped by zenith angle configuration (15°, 30° and 54.7°). Different colors denote different scan techniques (VAD 6, VAD 24 and DBS). Whiskers indicate the entire range of values, including the maximum and minimum. In addition, mean values are marked by a black diamond.

we are able to asses the reliability of the results we obtained from singular simulations.

We find that the flow regime affects the performance of lidar measurements. Convective conditions are generally characterized by heterogeneous flow structures, so that the requirements for wind vector retrieval with lidar scanning schemes are insufficiently met. Because the flow regime depends on meteorological parameters, such as stratification, geostrophic forcing, and near-surface heat flux, the lidar error is sensitive to these parameters. The results show that the lidar error increases in flow regimes where buoyancy driven structures prevail, whereas it decreases under shear driven conditions, as the flow field is more horizontally homogeneous. The error is comparatively small in a weakly convective boundary layer (RMSD below $0.4 \, \mathrm{m \, s^{-1}}$ for wind speed measurements averaged over 10 min) and large in strongly convective conditions (RMSD up to $3 \, \mathrm{m \, s^{-1}}$ for wind speed measurements averaged over 10 min).

We offer an estimate for the comprehensive error (RMSD) that can be expected for measurements within the boundary layer for different averaging intervals. Table 2 summarizes the results of all simulation cases for the 10-min averaging interval. We find that averaging measurements over periods longer than 10 min considerably reduces the error. For the moderately convective cases, mean errors for measurements averaged over a 30-min period range between $0.1 \, \mathrm{m \, s^{-1}}$ and $0.3 \, \mathrm{m \, s^{-1}}$. In





a purely convective situation we find errors up to $1\,\mathrm{m\,s^{-1}}$. Extending the averaging interval to 60 min generally reduces the error further, such that mean errors for wind speed do not exceed $0.8\,\mathrm{m\,s^{-1}}$. However, averaging intervals of 60 min might be

too long for practical application, as atmospheric conditions are seldom stationary in reality. The averaging interval of 30 min, which is common in meteorology, is therefore recommended. Note that homogeneity of the wind field may not be possible to achieve in very convective conditions, even with long averaging periods and that thus a certain error is to be expected for such measurements.

The study also shows that errors for wind speed measurements in convective situations decrease for larger zenith angle

configurations. Of the three zenith angles examined, the 54.7° angle configuration yields the smallest errors for wind speed, regardless of the scanning scheme used. This result contradicts suggestions by Courtney et al. (2008) and Klaas (2020) who argued that a smaller zenith angle reduces the lidar error, because the measurement points are closer together. Intuitively, it could be assumed that, in this case, the wind field across the area spanned by the measurement points is more likely to be homogeneous. Conversely, the results of this study suggest the opposite. This can possibly be explained by the larger contribution

of horizontal wind components on the radial wind measurement for larger zenith angle configurations. It appears that this effect outweighs the higher probability of a more inhomogeneous wind distribution within the larger probe volume. These findings support results by Hofsäß et al. (2018) who found measurements, performed during a field experiment, resulted in smaller errors, when a larger zenith angle was used. Referring to Teschke and Lehmann (2017), the effects of error propagation when deriving the horizontal wind vector from slanted radial wind measurements could also provide an explanation for this result. It

remains, however, to proof whether this finding also applies to other boundary layer regimes (e.g., a shallower (deeper) ABL with different typical sizes of the dominant convective structures).

The advantage of measurements with a large zenith angle over measurements with a small zenith angle is particularly noticeable in the strongly convective case. Here, errors can be reduced roughly by two thirds when using a zenith angle of 54.7°. In the moderately convective cases, this effect is also observed, although not quite as pronounced. Contrary to measurements of

the horizontal wind speed, measurements of the vertical wind exhibit smaller errors with smaller zenith angles, regardless of the meteorological situation. The results do not allow for a general statement as to how the zenith angle configuration affects the error in the wind direction.

The assessment of different scanning schemes for wind vector retrieval (VAD 6, VAD 24 and DBS) shows that no scanning scheme is generally superior. Instead, different combinations of scanning schemes and zenith angle configuration prove

favorable, depending on the meteorological situation, as well as the quantity to be measured. In the purely convective case the VAD schemes yield slightly smaller errors for wind speed than the DBS scheme. This also applies to measurements of the vertical wind, whereby the advantage of the VAD schemes over the DBS schemes is even more pronounced. Measurements of horizontal wind speed in the moderately convective case show best results with the DBS scheme, while the same is clearly inferior when measuring vertical wind. Notably poor performance of horizontal wind speed measurements is found for the

VAD 24 scheme in conjunction with a 15° zenith angle configuration.

In most of the analyzed cases, VAD 6 proves superior to VAD 24 for horizontal wind speed measurements. Exceptions are the purely convective case, where RMSD for both scan schemes are almost equally large, and the moderately convective cases,





**Table 2.** Overview of results for lidar measurements in different meteorological conditions averaged over a period of 10 min. Shown are lidar errors (RMSD in $\mathrm{m\,s^{-1}}$ over the extent of the respective boundary layer height) for different zenith angle configurations and scan schemes. Case numbers are according to Table 1. The cases are sorted by atmospheric stability, beginning with the least stable case. Values for case 4 and case 1 are mean values of the ensemble.

| Case | $\phi = 15°$ | | | $\phi = 30°$ | | | $\phi = 54.7°$ | | |
|---|---|---|---|---|---|---|---|---|---|
| | VAD 6 | VAD 24 | DBS | VAD 6 | VAD 24 | DBS | VAD 6 | VAD 24 | DBS |
| 4 | 1.07 | 1.07 | 1.18 | 0.63 | 0.61 | 0.72 | 0.36 | 0.34 | 0.40 |
| 3 | 0.36 | 0.42 | 0.40 | – | – | – | – | – | – |
| 1 | 0.29 | 0.52 | 0.27 | 0.21 | 0.25 | 0.19 | 0.23 | 0.2 | 0.19 |
| 2 | 0.14 | 0.38 | 0.13 | – | – | – | – | – | – |

were VAD 24 performs slightly better than VAD 6 when a zenith angle of 54.7° is used. In all other cases, VAD 6 yields considerably smaller errors than VAD 24, in particular for short averaging times (10 min). It appears that the better statistical averaging over 20 measurements along each scan direction within 10 min for VAD 6, compared to 5 measurements for VAD 24 can improve the quality of wind vector retrieval.

As for the wind direction, results suggest that the DBS scheme tends to yield the smallest errors. Furthermore, we find that orientation of the instrument with respect to main wind direction has negligible effect on measurement accuracy.

In summary, the virtual measurements performed in this study show that wind profile retrieval with Doppler lidars is indeed negatively affected by turbulent fluctuations of the wind field, which confirms results of previous studies on this topic (Lundquist et al., 2015; Gasch et al., 2020). We find that the retrieval error is related to the flow regime and the associated coherent structures. The instrument configurations as well as the scanning scheme significantly influence the lidar error in convective conditions and must therefore be chosen carefully, depending on the quantity to be measured. In very convective conditions, extending the averaging period to more than 10 min additionally mitigates the error. The uncertainties found in this study are within the range of deviations found in inter-comparison studies between wind lidar and other wind measurement techniques (Smith et al., 2006; Kindler et al., 2007; Päschke et al., 2015). Doppler lidar based wind data can thus be assumed equally well-suited for assimilation into NWP models.

In this study we focused on the adverse impact of flow heterogeneities on the wind retrieval, while neglecting other possible sources for errors and uncertainties (such as inaccuracies due to range gate averaging or uncertainty of radial velocity measurements). The absolute error values presented here should therefore be viewed with caution, since they do not account for such effects.

Furthermore, some of the simulation cases presented in this study are only individual example simulations. The results generated with these simulations merely give a rough idea about the tendency of the expected error and should be considered with





reservation. For two selected situations we performed ensemble simulations. Although these ensembles consisted of only ten members (a number certainly too small to obtain statistically robust results) they can provide a more profound error estimation for practical applications. Moreover, some general dependencies of the wind retrieval errors on averaging time, zenith angle, and ABL forcing regime become obvious. It is, however, recommended to perform similar future studies with a larger ensemble for each of the simulation runs.

To quantify the lidar error, we chose the RMSD, calculated as a bulk value over the vertical extent of the boundary layer. While this error metric serves as a condensed estimate of the expected error it fails to provide information on the errors at a given height range. Since the diameter of the scan circle increases with height, it can be expected that deviations from truth above the lidar location increase simultaneously, at least in the lower and middle ABL. Closer to the ABL top deviations are expected to decrease again, since the wind will approach the geostrophic value there. For differentiated statements about the 480 error behavior in relation to height above ground, a more detailed quantification of the lidar deviation, in addition to the RMSD, is desirable.

The matter of optimal zenith angle configuration for lidar wind profiling has been debated controversially and different conclusions have been reached (Courtney et al., 2008; Bingöl et al., 2008; Hofsäß et al., 2018; Klaas, 2020). The results of this study suggest that lidar configurations using a zenith angle of 54.7° yield most accurate results for measurements of horizontal 485 wind speed in turbulent flow conditions. Further investigation of this topic is desirable. We suggest a systematic analysis of the error behavior as a function of boundary layer depth, as this could provide meaningful clues as to which role coherent structures play with regard to zenith angle width. In addition, a measurement campaign to validate the findings from the simulations would be ideal, preferably using multiple instruments simultaneously to test various scan techniques and zenith angle configurations. This could help to assess the transferability of this study's results to practical application.

Virtual measurements implemented in LES have proven as a valuable tool to validate Doppler-lidar wind profiling. One major advantage of this tool is seen in the fact, that LES allow for the derivation of consistent wind profiles across the entire depth of the ABL, while standard measurement systems such as tower, sodar, radar wind profiler all are subject to limitations in range and resolution and partly rely on similar scanning assumptions as the Doppler lidar. With LES it is also possible to extend the measure of reference for the accuracy of measurements. For example, one could consider the mean over the lidar 495 scan circle, or the horizontal mean profile over a certain area. This would provide an indication of spatial representativeness of measurements, as well as allow for an evaluation of the significance of scan strategy in relation to spatial variability of the wind field.

Another perspective for future LES-based investigations could be to include land surface heterogeneity and even orographically structured terrain into the simulations. These aspects affect the flow and lead to more complex flow structures (such 500 as bending when flowing over a hill, or flow separation (Finnigan, 1988; Grant et al., 2015) that are difficult to capture with measurement systems (Bradley et al., 2015). The problem of lidar measurements over complex terrain has been investigated experimentally, as well as with flow models (Bingöl et al., 2009a, b; Pauscher et al., 2016; Klaas, 2020), but no study using LES in conjunction with a virtual lidar has been conducted so far. The ability of LES to represent such flow structures makes this technique a valuable tool for validation of measurement strategies under challenging flow conditions. In conclusion, we



are convinced that virtual measurements hold great potential with regard to quantifying measurement uncertainty and propose that their area of application be further extended.

*Data availability.* Data and code used for the presented research may be provided by the authors upon request. For this purpose please contact Charlotte Rahlves (rahlves@muk.uni-hannover.de).

*Author contributions.* Frank Beyrich and Siegfried Raasch conceived of the presented the ideas, designed the scientific questions and work
plan of the study and supervised the work. Charlotte Rahlves implemented the virtual lidar code, performed the numerical simulations and analyzed the results. All authors discussed the results and shaped the research. Charlotte Rahlves wrote the manuscript with input and critical feedback from all authors.

*Competing interests.* The authors declare that they have no conflict of interest.

*Acknowledgements.* The authors would like to acknowledge Lennart Böske for the initial development of the virtual lidar for PALM as well
as for providing preceding analysis results. Equally, we would like to thank Katrin Gehrke for providing preceding results and scripts for analysis. The work was supported by the North-German Supercomputing Alliance (HLRN).

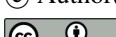



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
