# Peer review of "Scan strategies for wind profiling with Doppler lidar - An LES-based evaluation."

_Atmospheric Measurement Techniques, 2021_

## Author Response (AR1)

**Author's response to comments by referee 1**

**Response to general comments:**

*1. Referee comment: The implementation of the virtual lidar in the LES seems to be able to change the azimuth and elevation angle of the lidar beam instantaneously. Depending on the lidar type, the scanner head can have a significant travel time resulting in either shorter dwell times or the measurements covering an azimuth range instead of a single point. I believe this behavior could be added to the virtual lidar with either pauses between the sampling of the lidar beams or by averaging a distribution of lidar beams within an azimuth range. At the least, this aspect should be touched upon in the discussion.*

1. Author's response: Yes, it is correct that the scanner head of a lidar either moves continuously, resulting in a measurement over a range of azimuth instead of a measurement along a single azimuth direction, or spends time changing positions, resulting in a shorter dwell time. We added a remark that this way of operating the virtual lidar in the LES does not fully represent the scanning regime of a Doppler lidar in reality. However, one should be aware of the fact, that a lidar works with pulse repetition rates of several kHz, a sampling we will not be able to simulate. In the end the lidar provides one radial velocity measurement every 5 seconds. We think that averaging the LES output over 25 values within 5 seconds is a fair representation. We added this argumentation to the description of the virtual lidar scanning regime.

1. Author's changes: Line 163 ff. in the revised manuscript.

*2. Referee comment:  The methods sections suffers from fragmentation of information and some inconsistencies in the usage of terms (see specific comments for examples). It might benefit from further streamlining.*

2. Author's response: This is a good point. We considered this in our revision by consistently using the term 'reference value' instead of 'truth value'.

2.Author's changes: See author's response above (for example, line 198 f. in the revised manuscript).

**Response to specific comments:**

*Referee comment: Line 11: Instrument orientation could refer to both north alignment or horizontal leveling.*

Author's response: We added "with respect to the mean flow".

Author's changes: Line 12  in the revised manuscript.

*Referee comment: Line 13: The authors could consider to include the scan duration and time averaging aspect of the results into the abstract.*

Author's response: We added the following sentence to the abstract: "Furthermore, we find that extending the averaging interval length of lidar measurements reduces the error."

Author's changes: Line 14 ff. in the revised manuscript.

*Referee comment: Line 65: The abbreviation RMSD was only introduced in the abstract, but not the text.*

Author's response: Corrected.

Author's changes: Line 69 in the revised manuscript.

*Referee comment: Line 150 – 152: From the description I gather that the virtual LiDAR measures 5 seconds at the first position (alpha=0), then travels instantaneously to the second position (alpha=15) and measures the for the next 5 seconds. Wouldn't it be more realistic that the virtual LiDAR measures e.g. for 1 second at first position and then four seconds later the next measurement for 1 second is made at alpha=15? This way the travel time of scanner is accounted for, which results in less temporal averaging in reality.*

Author's response: See answer to general comment no. 1.

Author's changes: Line 163 ff. in the revised manuscript.

*Referee comment: Line 168: The same for averaging effects from the pulse length.*

Author's response: We fully agree that pulse-length averaging effects might be an additional source of error or uncertainty. However, it was not the aim of the present study to provide a full error analysis of Doppler-lidar wind measurement. Our focus was clearly on the errors introduced by different scan configurations, this was already explained in the text (now lines 182-184 in the revised manuscript). In fact, the pulse-averaging effect was investigated in a previous study by Gehrke (2019), and it was found to have a negligible effect when analyzing the errors of different scan regimes at a given height or averaged across the bulk of the atmospheric boundary layer.

Author's changes: None.

*Referee comment: Line 178: The height interval is always from 50 m to the top of the atmospheric boundary layer. To avoid fragmentation of information, this could be stated directly here.*

Author's response: Adapted.

Author's changes: Line 193 in the revised manuscript.

*Referee comment: Section 3 (and maybe throughout the manuscript in general): The reference values from the LES are referred to as "predicted values", "truth value", or "reference". The output of the virtual lidar are referred to as "observed values", "measured values", "virtual measurements", or "lidar values". I believe that settling on one specific term for each would increase clarity of the manuscript.*

Author's response: See answer to general comment no. 2.

Author's changes: See answer to general comment no. 2.

*Referee comment: Line 182: The description of the processing here is different to the description in line 154. The previous page states that the data is first accumulated in 120 s steps and later averaged again to 10/30/60 minutes. Here the description states that the individual beams are accumulated and then directly averaged 10/30/60 minutes.*

Author's response: We realize that the description of the processing might be contradictory to earlier descriptions of the scan procedure. We therefore rephrased this passage as well as the part in line 154. Note that the averaging process is described in line 160 – 164 (now line 173 to 177 in the revised manuscript).

Author's changes: Line 163 and 194 ff. in the revised manuscript.

*Referee comment: Line 186: Insert "the" before lidar.*

Author's response: We assume the reviewer was referring to line 187 "...measurement of the horizontal wind with lidar scan technique ...". We decided to leave this part unchanged, as we believe this is the grammatically correct form.

Author's changes: None.

*Referee comment: Line 190-192: Sometimes the wind profiles from a VAD scan are also interpreted to be representative for a spatial average across the scanning cone (in opposition to being representative for a column above the instrument). I believe this would be another aspect that could be investigated with the setup used here.*

Author's response: Yes, the reviewer made a very good point here. We did actually also investigate the spatial representativity of the measurements by comparing the measured profiles to the spatial average across the scanning cone, as well as to the average across the entire simulation domain. For this publication we decided to restrict ourselves to comparing the measurements to the vertical column for two reasons: Fist of all, as we stated in line 187 ff., one is usually interested in a wind profile above a certain location, for example when it comes to data assimilation into NWP models. Second, we feel that extending the discussion to additional reference wind profiles would overload this publication, which is already quite long.

Author's changes: None.

*Referee comment: Line 277: insert "the" before wind speed.*

Author's response: We left this in the original form since we feel that both versions are possible, and two lines later we use "wind direction" also without an article and the reviewer agreed on that.

Author's changes: None.

*Referee comment: Line 279: Did the authors use an arithmetic mean or an angular mean for the wind directions?*

Author's response: Here, the arithmetic mean was used.

Author's changes: None.

*Referee comment: Line 299-304: Was there a specific motivation why the north alignment of the Doppler lidar was investigated? To me this would fall into the same category as scanning the VAD in a counter-clockwise direction: I would expect some small random variations of the numbers, but no systematic difference. There is nothing wrong with this part, but it strikes me as an odd thing to investigate and was not well motivated in the introduction.*

Author's response: The motivation for including this investigation was that the orientation of the lidar could have an impact on the measurement result if quasi-stationary non-symmetric coherent structures are present in the boundary layer. We did not find this hypothesis confirmed, nevertheless, we believe it is a relevant result that should be included. To make our motivation for this more clear, we added a sentence in section 3.1.

Author's changes: Line 180 f. in the revised manuscript.

*Referee comment: Fig. 4 and Fig. 5: The panels for the 10-min average and 30-min average are unclear to me, because I only see one line for the reference / observation, but I would expect that there are 6 and 2 observations, respectively. Are those panel showing the mean RMSD?*

Author's response: In line 275 (in the revised version now line 289 ff.) we note that all RMSD values are mean values over a 60-min measurement period. Thus, there is only one single value for the 60-min interval, while for the 30 (10) -min interval, the value represents an average over two (six) values. We added a note to the figure caption to make this more clear for the reader. For the profile figures, such as Fig. 5, for reasons of

clarity, we display only one exemplary measurement period. We changed the text of the figure caption to "...profiles obtained over one exemplary 10-min period...".

Author's changes: Captions of Fig. 4 and Fig. 5.

*Referee comment: Line 331-332: To me it seems that identical values exist for top left, top center, and bottom left and remarkably close values for top right?*

Author's response: Corrected.

Author's changes: Line 348 f. in the revised manuscript.

*Referee comment: Line 498: Some of the limitations of this study like not accounting for range gate effects were already brought up in the methods section. Maybe the same could be done with the effect of surface heterogeneity on the error to put the readers mind at ease that it has not been forgotten.*

Author's response: We added a sentence to section 4, mentioning that all simulations are carried out over flat and homogeneous surface (line 210 f. in the revised manuscript).

Author's changes: Line 210 f. in the revised manuscript.

**Author's response to comments by referee 2**

**Response to specific comments and questions:**

**1. Variation of the retrieval error with altitude**

*Referee comment: The uncertainty for each retrieval configuration is given as a single value for the whole profile. How does the uncertainty vary with altitude, and particularly with scanning angle? The vertical profile of the uncertainty arising from the impact of turbulence is of clear interest, especially if there is a relationship between altitude and the size of the coherent turbulent structures being generated. Figure 2 shows that the larger scanning circles may match or exceed the coherent turbulent structure scales by 500 m altitude, but not at low altitudes (for this example). Figures 3 and 5 also suggest that the impact may vary with altitude substantially, particularly with respect to the zenith angle used. Since the profiles have already been generated, including such results (and adding a figure or two) would greatly enhance the impact of these results.*

*Quantifying any vertical variation in the uncertainty is important because a major goal of the wind profiling community is to provide an estimate of the overall retrieval uncertainty (i.e. combine these uncertainties with the instrument measurement uncertainties, which are likely to vary with range).*

Author's response:  The reviewer made a very good point here, it is absolutely true that knowledge of the error behavior with respect to altitude is essential for operational use and we agree that investigating this is the next step to take. We did actually start investigating the variation of the retrieval error with altitude. We did, however, come to the conclusion that this is an aspect that needs more thorough analysis, which we believe should be done on the basis of comprehensive ensemble simulations. We also believe that the manuscript, as it is, is already quite long and we feel that including a comprehensive analysis of the retrieval error variation with altitude would overboard this study. However, we decided to add a first exemplary analysis on this aspect as an appendix.

Author's changes: Appendix in the revised manuscript (line 529 ff.).

**2.  Averaging procedure**

*Referee comment: This study generates the averaged wind profiles by averaging the radial measurements first from all scans within the averaging period, before the wind retrieval is performed. What is the impact if the wind retrieval is performed per scan, and the retrieved wind profiles then averaged (in vector form)? Does this make a*

*difference to the conclusions? This may be important in operational situations where rapid update is requested (e.g. a wind retrieval is provided within five minutes for aviation forecasters or for data assimilation); waiting for 30 minutes may not be an option for these users.*

Author's response: This is two questions in one: (1) Changing the sequence of averaging vs. wind retrieval and (2) rapid output of mean values.

(1) For the averaging procedure, we followed the procedure most commonly applied in operational use.

(2) The averaging procedure allows for different averaging intervals to be taken into account simultaneously. The lidar completes multiple revolutions. The number of revolutions that are used for averaging depends on the desired averaging time. Therefore, a 10-min average can be obtained, while later the 30-min average can be obtained, as well.

Author's changes: None.

**3. Instrument measurement uncertainty**
*Referee comment: The difference in performance between the 6-beam VAD and 24-beam VAD is a surprising result, although the authors do provide a reasonable suggestion for why this is the case. As clearly stated in the manuscript, this study was only intended to quantify the impact of turbulence on wind retrievals and not to attempt to create a fully-featured Doppler lidar simulator. However, all retrievals presented here implicitly assume that the radial velocity measurement has no instrument measurement uncertainty. What would be the impact on the retrievals if a random uncertainty with a standard deviation of 0.1 m s-1 (for example) was added to each radial velocity?*

Author's response: As the reviewer said and acknowledged, it was not the aim to built a fully-featured Doppler-lidar simulator or to provide a fully comprehensive error analysis. We explicitly focused on the scanning strategies and the measurement uncertainty arising due to the violation of the homogeneity assumption. To add a measurement uncertainty would go beyond the scope of the study and overload the manuscript.

Author's changes: None.

**Response to technical comments:**

*Referee comment: Line 2: Suggest using 'Doppler lidar scanning' rather than 'Lidar scan'.*

Author's response: Adapted.

Author's changes: Line 2 in the revised manuscript.

*Referee comment: Line 5: Suggest 'virtual Doppler lidar measurements' in place of 'virtual measurements'.*

Author's response: Adapted.

Author's changes: Line 5 f. in the revised manuscript.

*Referee comment: Line 14: Suggest 'measurable impact' instead of 'relevant effect'.*

Author's response: Adapted.

Author's changes: Line 16 in the revised manuscript.

*Referee comment: Line 16: Suggest 'monitoring air quality' instead of 'air quality control'.*

Author's response: Adapted.

Author's changes: Line 20 in the revised manuscript.

*Referee comment: Lines 21-22: Do you mean 'Long-range scanning Doppler lidar'? Not all Doppler lidars have this range. Is vertical resolution an important part of this statement? If so, give a typical resolution. Do you mean 'nearest 100 m in range' as the minimum height depends on both the minimum range and the choice of elevation angle.*

Author's response: We have modified the text as follows: "Profiling lidars can cover almost the entire vertical extent of the ABL with a reasonably high vertical resolution,

except for the lowest 50 m to 100 m, depending on the scan elevation angle." (now line 24-25 in the revised manuscript)

Author's changes: Line 24 f. in the revised manuscript.

*Referee comment: Line 26: Replace 'to expected' with 'to be expected'.*

Author's response: Corrected.

Author's changes: Line 30 in the revised manuscript.

*Referee comment: Line 36: Do you mean the scales of turbulent motion relevant for Doppler lidar? This may be true for DNS but for LES this would depend on the both the temporal and spatial configuration.*

Author's response: Yes, we mean the turbulent motion relevant with respect to Doppler-lidar. Of course this depends on the temporal and spatial separation, but here we say that in principle LES can do this job and we think that our set-up (with dx = 5 m and dt = 0.2 s) matches these goals.

Author's changes: Line 40 in the revised manuscript.

*Referee comment: Line 40: Replace 'like' with 'as'.*

Author's response: Corrected.

Author's changes: Line 45 in the revised manuscript.

*Referee comment: Line 41: Do you mean one scan sequence - i.e one VAD scan at one elevation?*

Author's response: In recent studies usually only one specific zenith angle configuration and/or one scanning scheme (VAD or DBS) has been investigated.

Author's changes: None.

*Referee comment: Line 44: Replace 'Those' with 'These'. Be clear here that you do not include the radial measurement (instrument) error.*

Author's response: Corrected.

Author's changes: Line 48 ff. in the revised manuscript.

*Referee comment: Line 49: Replace 'such features as cellular structures, streaks or roll convection is' with 'features such as cellular structures, streaks or roll convection are'.*

Author's response: Corrected.

Author's changes: Line 54 in the revised manuscript.

*Referee comment: Line 64: Replace 'reference' with 'the reference profile in the LES simulation'.*

Author's response: Adapted.

Author's changes: Line 68 in the revised manuscript.

*Referee comment: Line 67: Replace 'confidence of' with 'confidence in'.*

Author's response: Corrected.

Author's changes: Line 72 in the revised manuscript.

*Referee comment: Lines 85-86: Suggest stating 'the vertical profile of the horizontal wind' rather than 'vertical wind profiles' here and elsewhere in the manuscript (e.g line 88).*

Author's response: Since it is not only the vertical profile of the horizontal wind, but rather the vertical profile of the horizontal, as well as, the vertical wind, we decided to use "the vertical profile of the three-dimensional wind vector" instead.

Author's changes: For example, line 91 and line 94 f. in the revised manuscript.

*Referee comment: Line 93: Is it strictly necessary for the beam to rotate clockwise? Can state that the beam is then rotated in azimuth.*

Author's response: Adapted.

Author's changes: Line 98 f. in the revised manuscript.

*Referee comment: Line 103: Italic 'n'.*

Author's response: Corrected.

Author's changes: Line 109 in the revised manuscript.

*Referee comment: Line 114-115: Is this always the case? Could sample N-E-S-W for example (less time spent scanning), or sample all 4 simultaneously. Also note that this describes only one version of the DBS scan, there are also 3-beam and five-beam variants.*

Author's response: Yes, the reviewer is right, this is one possible realization of a classical DBS scan and others with 3 or 5 beams are frequently used as well. We did add a note of this to the manuscript. And yes, N-E-S-W would need less time for a scan, however, to closer meet the stationarity assumptions inherently made in eq. (5)-(7) we decided to probe the opposite azimuth directions one after the other. Sampling the four beam simultaneously would not apply for a commercial lidar.

Author's changes: Line 119 f. and line 122 ff. in the revised manuscript.

*Referee comment: Line 122: Should u, v, w, denote the wind vectors rather than the components?*

Author's response: We changed this to "the three-dimensional wind vector".

Author's changes: Line 131 in the revised manuscript.

*Referee comment: Line 183: Replace 'truth' with 'reference'.*

Author's response: Adapted.

Author's changes: Line 198 f. in the revised manuscript.

*Referee comment: Line 196: Replace 'buoyancy to shear driven' with 'buoyancy-driven to shear-driven'.*

Author's response: Corrected.

Author's changes: Line 212 in the revised manuscript.

*Referee comment: Line 199: Replace 'literature' with 'the literature'.*

Author's response: Corrected.

Author's changes: Line 214 in the revised manuscript.

*Referee comment: Line 237: Replace 'until a turbulent flow' with 'until turbulent flow'.*

Author's response: Corrected.

Author's changes: Line 253 in the revised manuscript.

*Referee comment: Line 240: Replace 'domain averaged resolved-scale turbulence' with 'domain-averaged resolved-scale turbulent'.*

Author's response: Corrected.

Author's changes: Line 256 in the revised manuscript.

*Referee comment: Line 484: Only out of the zenith angles studied in the manuscript!*

Author's response: We changed the respective passage to: "The results of this study suggest that out of the zenith angle configurations investigated in this study, lidar configurations using a zenith angle of 54.7° yield most accurate results for measurements of horizontal wind speed in turbulent flow conditions."

Author's changes: Line 502 ff. in the revised manuscript.